# High-resolution positron emission microscopy of patient-derived tumor organoids

Syamantak Khan [1,5], June Ho Shin[2,5], Valentina Ferri [3], Ning Cheng[4], Julia E. Noel[2], Calvin Kuo [4], John B. Sunwoo [2] & Guillem Pratx [1✉]

Tumor organoids offer new opportunities for translational cancer research, but unlike animal models, their broader use is hindered by the lack of clinically relevant imaging endpoints. Here, we present a positron-emission microscopy method for imaging clinical radiotracers in patient-derived tumor organoids with spatial resolution 100-fold better than clinical positron emission tomography (PET). Using this method, we quantify [18]F-fluorodeoxyglucose influx to show that patient-derived tumor organoids recapitulate the glycolytic activity of the tumor of origin, and thus, could be used to predict therapeutic response in vitro. Similarly, we measure sodium-iodine symporter activity using [99m]Tc- pertechnetate and find that the iodine uptake pathway is functionally conserved in organoids derived from thyroid carcinomas. In conclusion, organoids can be imaged using clinical radiotracers, which opens new possibilities for identifying promising drug candidates and radiotracers, personalizing treatment regimens, and incorporating clinical imaging biomarkers in organoid-based co-clinical trials.

[1] Department of Radiation Oncology, Division of Medical Physics, Stanford University School of Medicine, Stanford, USA. [2] Department of Otolaryngology, Division of Head and Neck Surgery, Stanford University School of Medicine, Stanford, CA, USA. [3] Department of Radiology, Division of Nuclear Medicine and Molecular Imaging, Stanford University School of Medicine, Stanford, CA, USA. [4] Division of Hematology, Department of Medicine, Stanford University School of Medicine, Stanford, CA, USA. [5] These authors contributed equally: Syamantak Khan, June Ho Shin. ✉email: pratx@stanford.edu

Patient-derived tumor organoids are miniature, three-dimensional, self-organized tissue culture models that are derived from primary patient tumor cells and studied in the laboratory[1,2]. These organoid cultures closely recapitulate the genetic and morphological heterogeneity, stromal components, and microenvironment of the original tumor[1–10]. In addition, they contain the same cancer mutations and genetic variations that are present in the patient of origin, and thus they can assist in the selection of individualized treatment, especially for patients who fail to respond to the first-line therapy[3,11,12]. In addition, panels of organoids derived from patient cohorts capture inter-patient variability of tumor phenotype for drug screening of common and rare cancers[8,13,14]. Organoid cultures have a fast turnaround time, higher throughput, high level of control at low cost, and thus offer an alternative to patient-derived xenografts for certain applications.

Although many studies have employed tumor organoids to evaluate different therapies, relatively little work has been done in the area of diagnostic imaging. One significant limitation, of course, is that the resolving power of common radiological tools is not suited to the miniature size of organoids. The ability to image organoids using clinical contrast agents with high spatial resolution would facilitate the rapid translation of biomedical information, as the same noninvasive biomarkers could be used in preclinical organoid studies and clinical trials. Organoids could also be used as a platform to develop novel clinical contrast agents. This very concept, 15 years ago, drove the development of miniaturized small-animal scanners, which are now in widespread use at academic research centers and in the laboratories of pharmaceutical companies[15,16]. However, owing to the 1000-fold difference in scale, few noninvasive imaging biomarkers are available that are applicable to both organoids and patients. Several biological end-points are used for organoid studies, but these bear little relevance to clinical biomarkers. For instance, organoid response to treatment can be assessed using colony formation assays[17], viability assays[18], and optical redox imaging[19,20], all of which are far removed from the standard clinical workflow. Likewise, radiological criteria such as RECIST (Response Evaluation Criteria in Solid Tumors)[21] are challenging to apply to organoid cultures.

The gold standard in cancer clinics for molecular assessment of normal and diseased tissues is positron emission tomography (PET), which is often combined with computed tomography (CT) to achieve multimodal noninvasive imaging of disease in vivo. The discovery of the Warburg effect has led to the extensive use of $^{18}$F-fluorodeoxyglucose (FDG) as the main PET tracer to map glucose metabolism in primary and metastatic tumors[22]. In the last decade, FDG-PET has been shown to drastically improve the diagnosis, staging, and subsequent treatment of cancers, and it has emerged as a reliable tool for assessing treatment response after chemotherapy and radiotherapy. Yet, few studies—if any—have investigated the potential of FDG as a biomarker for assessing the response of patient-derived tumor organoids to therapy.

Although the low resolution of clinical PET (~4–8 mm) is a significant limitation for organoid imaging, recent work has shown the feasibility of imaging PET tracers in 2D cell cultures with single-cell resolution using an approach known as radio-luminescence microscopy (RLM). RLM uses a scintillator crystal to convert beta radiation emanating from the cells into optical flashes detectable in a single-photon-sensitive microscope, and thus provide high-resolution 2D imaging of radionuclide distribution in live cells[23,24].

This study presents a proof of principle for imaging organoids using the RLM technique. We call this specific method of imaging Positron-Emission Microscopy of Organoids (oPEM) to highlight its relevance to clinical PET imaging. In combination with brightfield (BF) and fluorescence microscopy, oPEM allows multimodal imaging of organoids with ~100-fold higher resolution than clinical PET, using both fluorescent probes and clinically relevant PET radiotracers. To investigate this approach, organoids were grown for 2–3 weeks in 3D hydrogel from tumor tissues derived from patients (Supplementary Table 1), then treated with $^{18}$F-FDG and placed on transparent CdWO$_4$ scintillator plates for imaging with RLM (Fig. 1). The spatial distribution of FDG within organoids was superimposed onto standard brightfield and fluorescent micrographs to identify and characterize areas of high glycolysis. The level of FDG

**$^{18}$F-FDG Positron Emission Tomography (FDG-PET)**　　　　**$^{18}$F-FDG Positron Emission Microscopy (FDG-oPEM)**

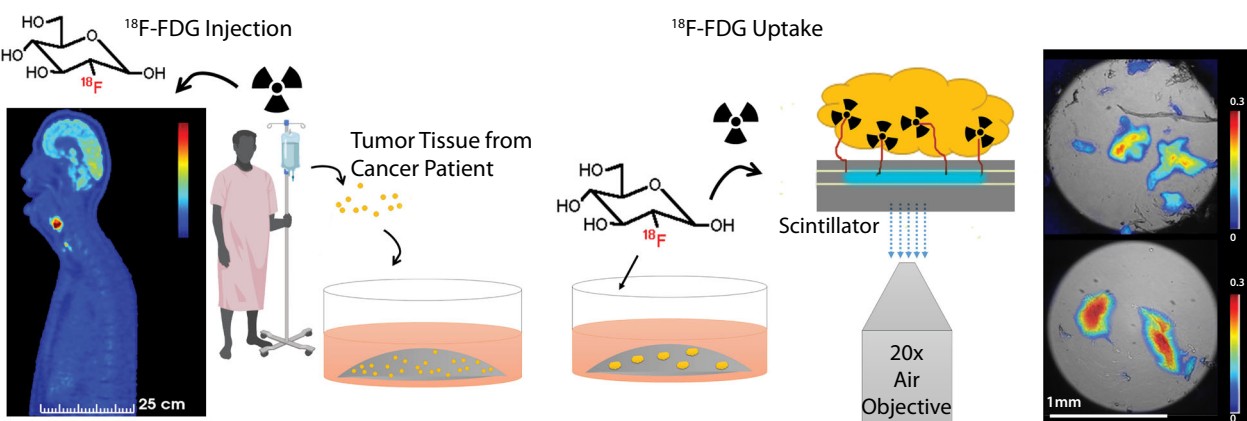

Submerged 3D Organoid Culture in basement membrane matrix  (2-3 weeks)

**Fig. 1 Schematic representation of the oPEM workflow.** Tumor organoids were cultured for 2–3 weeks in a submerged basement membrane matrix culture system. Patient-derived organoids (right) were imaged with high resolution using FDG, a radiotracer commonly used in the clinic for diagnosis and staging of head-and-neck cancer patients (left). The organoids were placed on thin inorganic scintillators and the resulting scintillation light was imaged with a highly sensitive microscope through a ×20 objective lens. The scale bars of PET and oPEM images highlight the large difference in image resolution. The color bar shows radioactivity (Bq/pixel) inside tumor organoids.

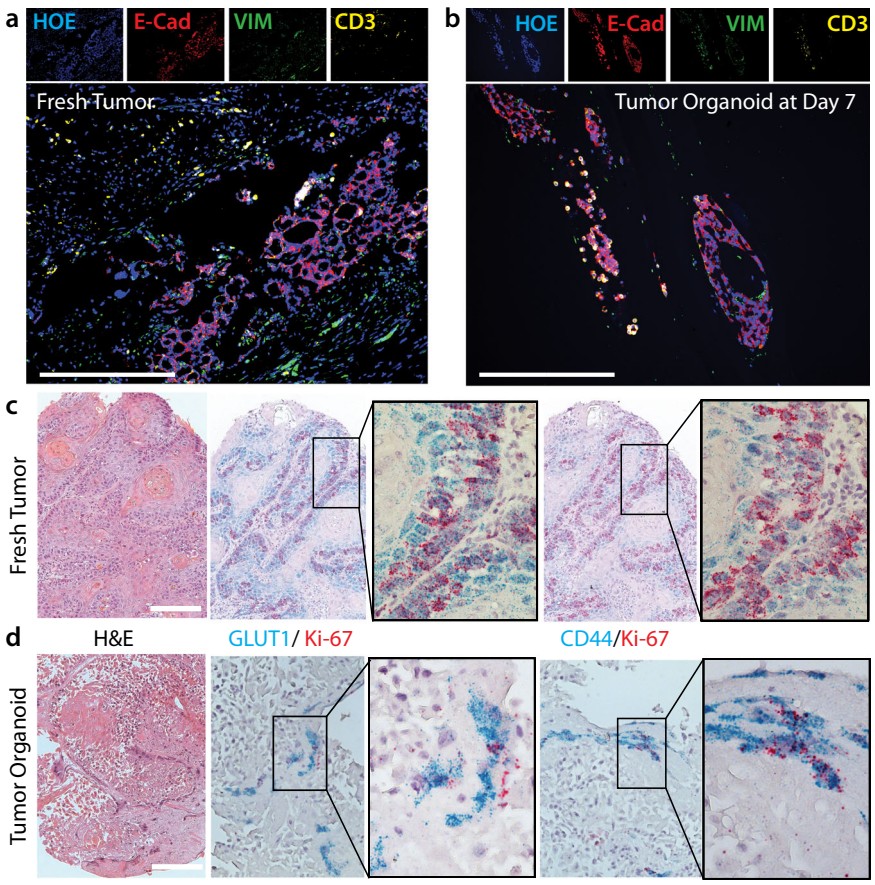

**Fig. 2 Characterization of head and neck tumor organoids grown from tumor tissue. a, b** Fluorescence immunohistochemistry comparing (**a**) fresh tumor sample and (**b**) tumor organoid grown from head-and-neck adenoid cystic carcinoma sample labeled with four markers: blue showing Hoechst 33342 (HOE, nuclei); red, E-cadherin (E-cad, tumor epithelial cells); green, vimentin (VIM, tumor-associated fibroblasts); and yellow, CD3 (tumor-infiltrating T cells). **c, d** Co-registration of hematoxylin & eosin staining (H&E) and RNA in situ hybridization (RNAscope duplex assay) detecting expression of the glucose transporter GLUT1 (blue), stem-cell marker CD44 (blue), and cell proliferation marker Ki67 (red) in **c** fresh human oral squamous cell carcinoma tissue and **d** corresponding tumor organoids. Scale bar: 0.2 mm.

uptake of individual organoids was then quantitatively compared with the uptake of the cells in the tumor of origin, as measured in vivo using FDG-PET. This analysis revealed considerable agreement between organoid uptake in vitro and tumor uptake in vivo. As a further demonstration of the utility of oPEM, organoids were grown from cisplatin-sensitive tumors, and their response to cisplatin was assessed according to the relative difference in FDG uptake. Finally, $^{99m}$Tc-pertechnetate uptake imaging of thyroid cancer organoids illustrates the applicability of oPEM beyond FDG.

## Results

**Tumor organoids recapitulate the microenvironment of the original tumor.** Techniques for generating high-quality tumor organoids have been developed and studied extensively for diverse cancers in the past years by the Kuo laboratory and others[5,25–27]. In this study, tumor organoids were seeded from processed surgical samples of head-and-neck cancer patients and cultured in basement membrane extract (BME), a soluble form of basement membrane purified from Engelbreth-Holm-Swarm tumor. A specialized culture medium (EN medium) containing Dulbecco's Modified Eagle Medium (DMEM)/F-12 supplemented with 10% Noggin-conditioned media, Nicotinamide (10 mM), N-acetylcysteine (1 mM), B-27 minus vitamin A (1×), Pen–Strep (1×), and EGF (50 ng/mL) was used to grow organoids.

These tumor organoids closely recapitulated the most salient features of the tumor of origin, as observed in previous studies[5,25–27]. After 1–3 weeks of culture, they displayed dysplastic epithelial features with keratin production and disorganized growth patterns, typical of squamous cell carcinoma (Supplementary Fig. 1). Fluorescence immunostaining of the organoid and the corresponding tumor of origin highlighted functional and cellular similarities between the two tissues (Fig. 2a, b). In both samples, E-Cadherin selectively identified the cancerous epithelial cells, Vimentin the tumor-associated fibroblasts, and CD3 the tumor-infiltrating T cells. The retention of stromal components after 7 days of in vitro culture is a crucial advantage of this cancer model, as treatment response is often influenced by a biologically active tumor stroma and its complex interactions with the cancer cells.

Furthermore, we performed RNA in situ hybridization of tumor and organoid samples to compare gene expression in matched organoid and tumor tissue. By comparing GLUT1 expression levels, we confirmed that the high expression of glucose transporters was conserved in tumor organoids and similar to that in the tumor of origin (Fig. 2c, d). Besides, CD44+ progenitor/stem-cell niches were identified in both tumors and organoids, highlighting the ability of both tissues to regenerate the original cancerous tissue. These experiments also stained simultaneously for Ki67, used as a proliferation marker. In summary,

the tumor organoids used in this study are effective and clinically relevant in vitro models for studying head-and-neck cancers.

**RLM enables imaging of glucose metabolism in tumor organoids**. In contrast to normal cells, cancer cells largely avoid the mitochondrial tricarboxylic acid cycle and instead rely on aerobic glycolysis for energy production. The advantage of aerobic glycolysis, also known as the Warburg effect, is still not entirely clear but the phenomenon is the molecular basis for assessing tumor burden in vivo using FDG-PET. Several studies using optical metabolic imaging have shown that tumor organoids display a glycolytic profile similar to that of solid tumors[10,28,29], however, the FDG uptake of tumor organoids has not yet been examined.

Using a single-photon-sensitive microscope (Supplementary Fig. 2), we imaged FDG uptake in organoids derived from clinical tissue specimens. Organoid cultures were incubated with FDG (1 mCi/ml) at 37 °C for 1–2 h, giving the radiotracer enough time to diffuse into the tissues, then washed for 30 min. The organoids were then gently dissociated from their matrix by slow micropipetting, transferred to a 0.5-mm-thick $CdWO_4$ scintillator, and imaged using RLM as previously described[23].

A few adjustments were made to the standard RLM protocol to allow thick 3D organoid samples to be imaged. RLM images are constructed from the radioluminescence flashes, which originate from individual positron decays occurring inside the organoid tumor. In one approach, called digital imaging, individual scintillation flashes collected by the electron-multiplying charge-coupled device (EMCCD) are precisely counted and converted to the number of radioactive decays to yield a composite map[30] (Supplementary Fig. 3). Although this digital approach is the method of choice for 2D cell imaging, it is less efficient for 3D tissue samples owing to the presence of a large number of cells within a small volume. Although problematic for digital imaging, the high count-rate creates enough scintillation signal for fast and direct analog measurements of the whole organoid using a single camera exposure of 10–300 s, no pixel binning, and an electron-multiplication gain of 600/1200. We further use a ×20 objective (NA = 0.75) to achieve an imaging field of view of 1.5 mm and an image pixel size of 3.2 μm.

For accurate quantification, we created a calibration curve from a known radioactivity distribution (Supplementary Fig. 4). The obtained calibration data matched the exponential decay curve of $^{18}F$, down to very low activity and camera signal. To verify the accuracy of this calibration method, we compared the radioactivity of organoids estimated from the images to the total radioactivity of individual organoids measured using a gamma counter.

To quantify radiotracer concentration in the oPEM images, EMCCD pixel counts were converted to radioactivity (Bq/pixel) using the slope of the calibration curve. A color map of the radioluminescence intensity depicts the distribution of FDG inside the tumor organoids and reveals spatial metabolic heterogeneity (Fig. 3a & Supplementary Fig. 5). The combination of brightfield and radioluminescence images plays a similar role as in clinical PET/CT, where FDG accumulation can be assigned to a specific region or tissue structure with high glucose metabolism.

A comparison with 2-(N-(7-Nitrobenz-2-oxa-1,3-diazol-4-yl) Amino)-2-deoxyglucose (2-NBDG), a fluorescent analog of glucose, revealed that FDG was more specifically taken up by tumor organoids, indicating the superiority of FDG over 2-NBDG to probe cellular metabolism (Fig. 3b). It is worth mentioning that, unlike FDG, any fluorescent analog of glucose would require a bulky fluorophore group attached to its carbon backbone. This requirement significantly alters the biochemical properties of the substrate, making it less reliable to study glucose metabolism in cells and tissues[31–33]. Besides, the fluorescent signal is sensitive to several extrinsic parameters such as polarity, molecular crowding, pH, and interaction with the environment resulting in unwanted fluorescence enhancement and quenching.

From this perspective, developing microscopic imaging techniques using FDG would be extremely useful for accurate imaging of glucose metabolism. Unlike other metabolic imaging approaches, FDG uptake has a clear biological meaning, as its uptake represents the total flux of glucose into glycolysis. In addition, compared with fluorescence imaging, the information obtained from FDG-based imaging is relevant to clinical practice and can be compared with in vivo glucose metabolism measured in tumors using FDG-PET. In addition to head-and-neck tumors, we investigated FDG uptake in organoid models of lung adenocarcinoma, renal cell carcinoma, and colorectal carcinoma. These organoids were imaged in situ using oPEM, without disrupting the Matrigel dome and the organoid microenvironment (Supplementary Fig. 6). In principle, the approach can be applied to any organoid model and any PET tracer for which there is no direct equivalent fluorescent analog available.

In another set of experiments, tumor organoids were co-labeled with live/dead fluorescent probes to assess the specific localization of FDG uptake (Fig. 3c). Generally, the radioluminescent hot spots were localized within the viable regions of the organoids. This positive correlation indicates that FDG uptake is specific and representative of metabolically active regions. No uptake was observed in non-viable organoid tissue, indicating that FDG is not retained unless metabolized by live cells. However, it should be mentioned that fluorescent live/dead probes are not expected to correlate exactly with FDG uptake, as their uptake is based on the difference of membrane permeability between live and dead cells instead of their metabolic state. Similarly, oPEM images can also be multiplexed with a range of other imaging techniques. For instance, histological sections were performed to show H&E (Fig. 3d) and RNA in situ hybridization (Fig. 3e) and were co-registered with FDG-oPEM images. Spatial registration can however be challenging given that oPEM images whole specimens, whereas other approaches require sectioning.

Finally, the spatial resolution of oPEM is crucial to accurately determine the spatial distribution of PET tracers (e.g., metabolic heterogeneity) inside complex tissue structures. Owing to the high specificity of the tracer uptake, the high resolution of oPEM could be advantageous to differentiate between cancerous and stromal tissues in these organoids. The full-width half-maximum of a tiny structure, measured using oPEM, was found to be 17 ± 1 μm (Supplementary Fig. 7). This result suggests that the resolving capacity in the current experimental conditions is very high when the source of the signal is very close to the scintillator surface. However, this value may overestimate the resolution of the entire image as the resolving power of oPEM degrades with increasing sample thickness. To estimate an average resolution over the whole image, we performed Fourier ring correlation (FRC) analysis by calculating the average cross-correlation between two images of the same sample along with concentric frequency domain rings (Supplementary Fig. 7)[34,35]. Using a preset threshold of 1/7, we estimated a limit of resolution of 60 ± 4 μm. To achieve even higher spatial information, organoids could be cut into thin sections and imaging could be performed digitally (as described before), although this invasive approach would prevent kinetic and longitudinal imaging. Nevertheless, FRC analysis suggests that oPEM of intact 3D organoids can still be performed with ~100 times higher resolving power than clinical PET imaging and ~20 times higher than microPET, which is used for preclinical imaging of small animals.

**Quantification of glucose metabolism inside organoid tumors**. The emerging use of patient-derived tumor organoids in

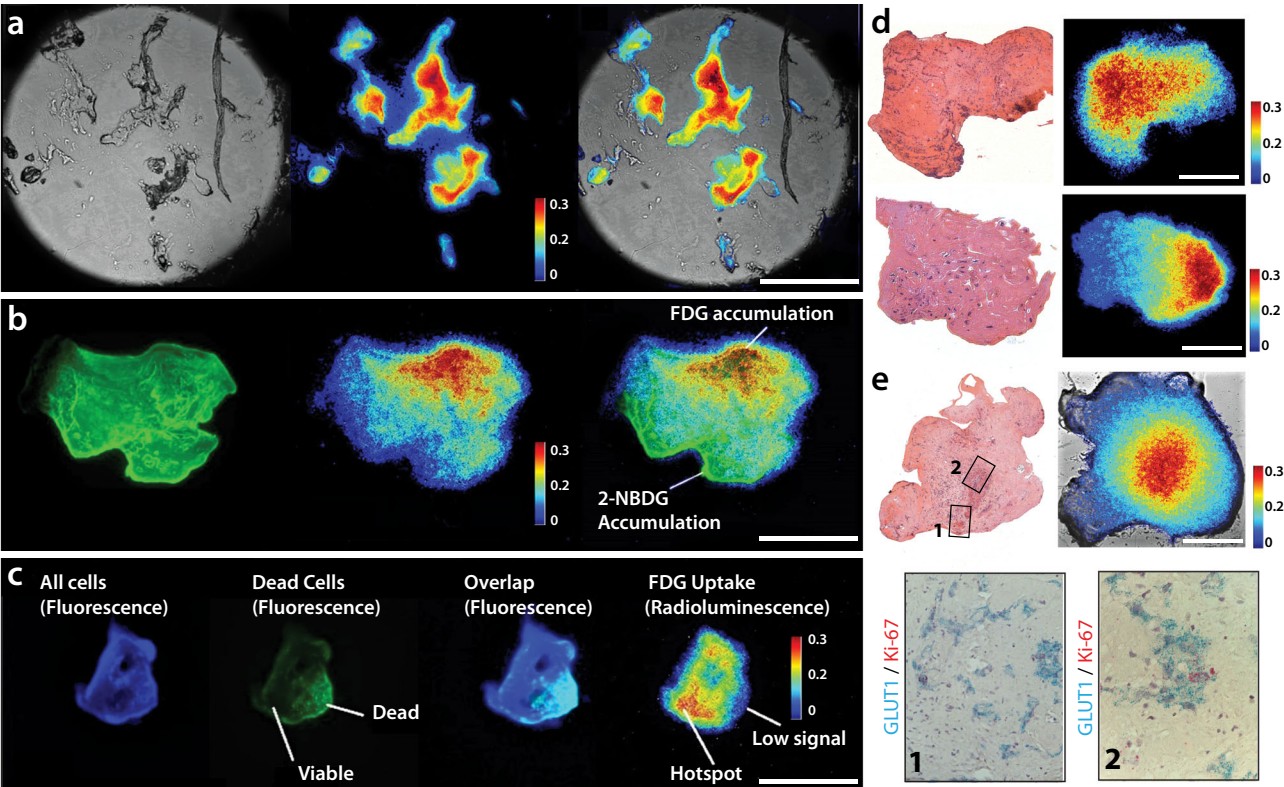

**Fig. 3 Positron emission microscopy of tumor organoids. a** Brightfield image (left), radioluminescence image (middle), and overlay (right) show the correlation between FDG uptake distribution and tissue structure. FDG uptake is elevated in most of the organoids. **b** Comparison between fluorescence imaging with 2-NBDG, a fluorescent glucose analog (left), and oPEM imaging with FDG (middle) indicates inconsistent co-localization (right) of the two probes. **c** Tumor organoid labeled with live/dead fluorescent stains (three left panels) shows that FDG uptake (right panel) is associated with tissue viability. Blue: Hoechst 33342 (all nuclei), Green: SYTOX green (dead nuclei). **d** FDG-oPEM of human oral squamous cell carcinoma organoids (whole mount) and co-registration with H&E staining of organoid sections. **e** Two different regions of an H&E section and its corresponding whole mount FDG-oPEM/BF (top panels) co-registered with in situ hybridizations (bottom panels) showing expression of the glucose transporter GLUT1 (blue) and cell proliferation marker Ki67 (red). The color bar shows radioactivity (Bq/pixel). Scale bar: 0.5 mm.

translational cancer research calls for new tools to produce reliable, reproducible, and quantifiable biomarkers that are consistent with clinical practice. We propose that oPEM may play such a role owing to its ability to image a variety of clinical radiopharmaceuticals. However, an important prior step is to demonstrate that oPEM images of organoid models are congruent with in vivo PET images of the tissue from which the organoids were derived. In the case of FDG, a key step in this analysis is to identify an intrinsic parameter that represents the metabolic state of the tissue, independent of the spatial dimensions of the system, the amount of FDG available to tumor cells, and the time course of the uptake. The standardized uptake value (SUV) is a well-established metric that accounts for the size of the patient and the injected tracer dose, but not for the kinetics of the tracer in the plasma compartment. As there is no tracer clearance for organoid cultures, the SUV may not be a suitable metric to compare organoids and in vivo tumor uptake.

This study considers instead the net uptake rate $K_i$, a metric derived from Patlak analysis assuming irreversible tracer uptake. The coefficient $K_i$ has units of inverse time and represents the net transport and trapping of FDG into tumor cells as a fraction of the known concentration of FDG in the plasma compartment. As virtually all clinical PET scans are performed as static scans, a simplified analysis was conducted. First, we assumed the y-intercept ($V_D$) to be negligible compared with the rate of tumor uptake. The approximate $K_i$ rate computed using this approach, sometimes referred to as the fractional uptake rate, is valid for

scans acquired at late time points. Second, in place of arterial blood sampling, we used a standardized input function derived from a population of over 101 patients[36]. The model provides an input function adjusted for patient height and weight. Third, we assumed the volume of blood in the region of interest to be negligible compared to the volume of tumor tissue. Using this model, we computed the $K_i$ of tumors from FDG-PET images, and that of organoids from oPEM and/or gamma counting measurements.

Generally, FDG uptake in the organoids was on the order of 10,000 kBq/ml, much higher than the uptake measured in tumors using PET (~10 kBq/ml). This apparent discrepancy arises owing to the difference in the concentration of FDG available to tumor cells. Although the dose of FDG used for organoid imaging (~30 MBq) was lower than the doses injected in patients (~300 MBq), the final concentration of the tracer was much higher in the organoids culture medium (~30,000 kBq/ml) than in the patient plasma (~30 kBq/ml) owing to the vastly different distribution volume for the two biological systems. It is interesting to notice that despite these differences, the calculated $K_i$ values fall in the same order of magnitude for both in vivo tumors and in vitro organoids (Supplementary Table 2). It should be mentioned here that the total molar concentration of FDG was kept <50 μM, significantly lower than the saturation level for the glucose pathway.

To test whether FDG-oPEM captures inter-patient variability, we cultured organoids from two patients who each presented with papillary thyroid carcinoma nodules of contrasting FDG avidity. The first patient (T1) presented with an FDG-avid thyroid nodule

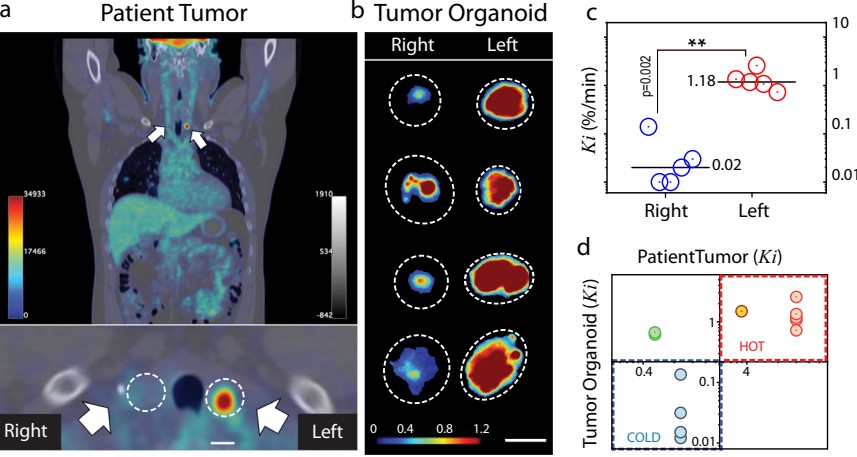

**Fig. 4 Comparison of PET vs oPEM imaging. a** PET/CT scan of patient T3. Two papillary thyroid carcinoma nodules (white arrows) show contrasting levels of FDG uptake. **b** oPEM images of tumor organoids derived from these nodules. The organoids retained the contrasting metabolic identity of the original two nodules. The dotted lines show the spatial extent of the individual organoids. The color bar shows radioactivity (Bq/pixel). Scale bar: 1 mm. **c** Scatter plot of FDG influx constant ($K_i$) for organoids ($n = 5$) derived from left and right nodules. The median uptake is shown as a black line. The organoids derived from the left nodule took up >10 times more FDG on average than those from the right nodule. An unpaired two-tailed $t$ test was applied for significance testing. **d** Scatter plot showing the correlation between the $K_i$ of organoids ($n = 13$) and tumors of origin ($n = 4$, shown in four different colors; Pearson's $r = 0.756$, $P = 0.0032$).

(SUV = 8.7; Supplementary Fig. 8), whereas the second patient (T2) had an FDG-cold nodule, with no visible uptake in the PET scan (SUV = 1.8; Supplementary Fig. 8). The difference in metabolic intensity between the two nodules was reproduced in vitro in organoids derived from the same tumors using oPEM. Organoids from the first patient showed higher FDG influx ($K_i = 1.1 \pm 0.3\%$/min) than those from the second patient ($K_i = 0.6 \pm 0.2\%$/min).

However, it could be argued that the difference in FDG activity may be influenced by differences in the genetic background and disease stage of the different patients. To overcome this confounding factor, we established organoids from a unique patient (T3) who presented with two thyroid nodules of similar size with contrasting FDG findings (Fig. 4a). The first nodule, located in the left lobe of the thyroid, had intense FDG uptake (SUV = 32.4) whereas the second nodule, located in the right lobe, was cold and not visible on the PET scan (SUV = 2.5; Supplementary Fig. 9). Both nodules were pathologically confirmed to be papillary thyroid carcinoma after resection. Thus, having two different tissue samples of the same disease from the same patient, we could test whether metabolic heterogeneity is conserved when tumor cells are grown as organoids.

From the PET scan, we calculated the FDG influx rate into the nodules and found it to be $K_i = 11.57\%$/min for the left side and $K_i = 0.89\%$/min for the right side. Similarly, oPEM imaging revealed a stark difference in FDG uptake between two groups of organoids (Fig. 4b). FDG uptake was visibly higher in all the organoids derived from the left nodule. However, in the other group, one of the organoids also presented relatively high uptake, potentially owing to heterogeneity in the tumor nodule used to seed these organoids. More quantitatively, gamma counting measurements show that organoids derived from the patient's left nodule had >10-fold higher FDG influx rate ($K_i = 1.4 \pm 0.3\%$/min) on average than those derived from the right nodule ($K_i = 0.04 \pm 0.02\%$/min; Fig. 4c). Overall, the data from these four tumors suggest that the bulk metabolic state of solid tumors is conserved when the tumors are grown in vitro as organoids (Fig. 4d); in other words, oPEM functionally recapitulates the original imaging findings from clinical PET studies.

**Assessing drug response in head and neck squamous cell carcinoma tumor organoids**. Given this association between

in vitro and in vivo metabolism, we sought to investigate the ability of oPEM to replicate, in an in vitro setting, the use of FDG-PET for monitoring therapeutic response in patients. The metabolic response of head-and-neck organoids was assessed using FDG-oPEM after cisplatin treatment (0, 1, and 10 μM for 24 hours; Fig. 5a & Supplementary Fig. 10). For clinical relevance, dosages of 1 & 10 μM were chosen as the IC50 value and local concentration of cisplatin in patient tumors often fall near that range[37]. FDG uptake shows a dose-dependent decrease in glucose metabolism after treatment with cisplatin. In quantitative terms, the uptake of FDG in these organoids was reduced >10-fold after treatment with 10 μM cisplatin. (Fig. 5b) with approximate FDG concentrations of ~3 MBq/ml and ~0.1 MBq/ml, before and after drug treatment, respectively. The significant decrease in metabolic response within 24 h of drug treatment reflects reasonable sensitivity of this approach implying, this method may identify cell fate early on the basis of cell glycolysis, which is the same reason why FDG-PET has proved to be more effective than anatomical measurements for monitoring therapeutic response in patients. Figure 5a also shows the live/dead fluorescent co-labeling, which was performed to identify the viable regions of the organoids. Again, the FDG uptake was mostly within the viable regions, indicating an inverse relationship between dead cell staining and FDG signal throughout the organoid structure (Fig. 5c). However, a perfect correlation between viability and FDG uptake was not expected as the dead cell stain does not label the weakly metabolic viable cells.

**Technetium scan of tumor organoids of papillary thyroid carcinoma**. Finally, to demonstrate the use of oPEM for tracers other than FDG, we investigated whether thyroid tumor organoids could recapitulate in vitro the avidity of thyroid carcinoma for iodine. In clinical oncology, radioiodine is widely used to diagnose and treat thyroid cancer. Uptake of iodine takes place across the membrane of thyroid follicular cells and thyroid cancer cells through an active transport process mediated by the sodium iodide symporter (NIS). To test whether NIS-dependent iodine uptake is conserved in organoids derived from iodine-avid papillary thyroid carcinomas, we used oPEM to measure the uptake of technetium-99m pertechnetate ($^{99m}TcO^-_4$), a

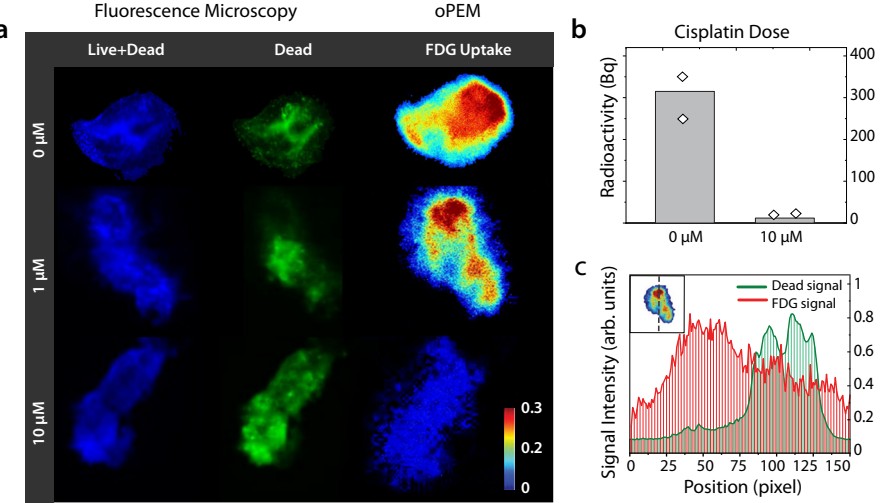

**Fig. 5 oPEM & fluorescence co-imaging of metabolic activity after cisplatin treatment. a** Untreated organoid shows a spatial pattern of cellular viability (fluorescence stain; left column) consistent with the FDG uptake profile (right column). Organoids treated with increasing doses of cisplatin experience a decrease in their metabolic activity. Blue: Hoechst 33342 (all nuclei), Green: SYTOX green (dead nuclei). The color bar shows radioactivity (Bq/pixel). **b** Quantitative FDG uptake inside tumor organoids treated with 0 or 10 μM dose of cisplatin ($n = 2$ organoids from two independent experiments). **c** The intensity profile of the FDG signal and green channel along the dotted line (inset) shows a spatial anti-correlation.

well-known NIS substrate[38]. The radiotracer $^{99m}TcO^-_4$ was used in this study because, unlike radioiodine, it is non-volatile and thus requires minimal safety precautions. We imaged whole organoids using 0.2 mm-thin cerium-doped gadolinium aluminum gallium garnet (GAGG:Ce) scintillator plates to detect the 140 keV internal conversion electrons (12% emission yield) emitted by $^{99m}Tc$ (Fig. 6a). Competitive inhibition of NIS using 1 mM potassium iodide (KI) resulted in a significant ~10-fold decrease in $^{99m}TcO^-_4$ uptake by papillary thyroid organoids, suggesting most of the uptake occurred through NIS transport (Fig. 6a, b). To further quantify the role of NIS, we performed a controlled study using MDA-MB-231 breast cancer cells transfected to express the NIS gene. These transformed cells rapidly took up $^{99m}TcO^-_4$ and reached kinetic equilibrium within 30–60 min. In contrast, wild-type MDA-MB-231 cells showed virtually no $^{99m}TcO^-_4$ uptake (Fig. 6c). Similar findings were observed using oPEM to image 3D multicellular spheroids made from these two cell lines (Fig. 6d).

## Discussion

In summary, a major finding of our study is that clinical PET tracers, such as FDG, can be imaged with high spatial resolution after they distribute in 3D organoid models of cancer. Using oPEM, we find that tumor organoids recapitulate specific metabolic features of the tumors of origin. The method can also monitor drug responses of organoid cultures. This concept is analogous to the use of FDG-PET for monitoring treatment response in cancer patients. The method is safe to use, cost-effective, relatively fast, and capable of screening samples with 100-fold higher resolution than a clinical PET scan. Thus, oPEM could be a useful tool for preclinical studies based on organoid models.

As a method, oPEM draws from our previous work developing RLM for imaging 2D cell cultures. The method was adapted to accommodate the specific characteristics of organoids, including the higher density of cells present in the field of view and the increased thickness of the sample. Our study found that analog imaging using relatively short exposure of a few minutes enabled fast imaging of large organoids, with only a moderate loss of spatial resolution compared to 2D cell imaging[39]. Owing to the

physics of positron propagation in tissue, oPEM allows effective imaging of any biological sample kept close enough (<100 μm) to the scintillator surface. In this way, RLM can perform the equivalent of a PET scan on patient-derived organoids to reveal the heterogeneous metabolic state of the tissue. Non-viable tissue does not retain FDG, indicating that FDG uptake is a specific surrogate for the flux of glucose into cells. In addition, our analysis of a small data set of matched PET/organoid scans suggests that the metabolic state of tumors is preserved when these tumor cells are grown in vitro as organoids. Accordingly, in the future, organoids could be grown from freshly excised tumor tissue and assembled into panels for testing novel therapeutics or radio-pharmaceuticals. They could also be used to predict the response of individual patients to therapy.

The use of organoid models is expanding rapidly, driven by factors such as cost, throughput, robustness, and versatility. Many of the hallmarks of cancer—metabolic reprogramming, drug resistance, and even metastasis—can be modeled using organoid models[29,40]. In this context, the use of oPEM would be beneficial for studies that are challenging to perform in vivo. For instance, oPEM could help elucidate the onset of metabolic reprogramming during tumorigenesis in normal tissue organoids[41,42], which could have value for understanding the feasibility of detecting early-stage tumors with PET. Moreover, oPEM could be used in areas beyond oncology to image normal tissue organoids and to study specific disease pathologies[4,42,43]. PET tracers are being developed for a wide range of diseases, ranging from neurological disorders to cardiovascular and infectious diseases. Organoids will provide a versatile platform to accelerate the development of improved PET tracers for these emerging applications.

In the clinic, PET does not predict which treatments are most likely to elicit a response; rather, the information is obtained post hoc, weeks after the treatment has started, thus delaying the switch to more effective therapy. There is an urgent need for approaches that can identify suitable treatments, especially for patients who fail first-line therapy. A number of recent studies support the use of tumor organoid models to predict the response of chemotherapy, combinational therapy, or immunotherapy responses of individual patients suffering from various cancers[6,40,44,45]. Thus, in this context, oPEM could become a valuable tool to predict therapeutic responses in a noninvasive

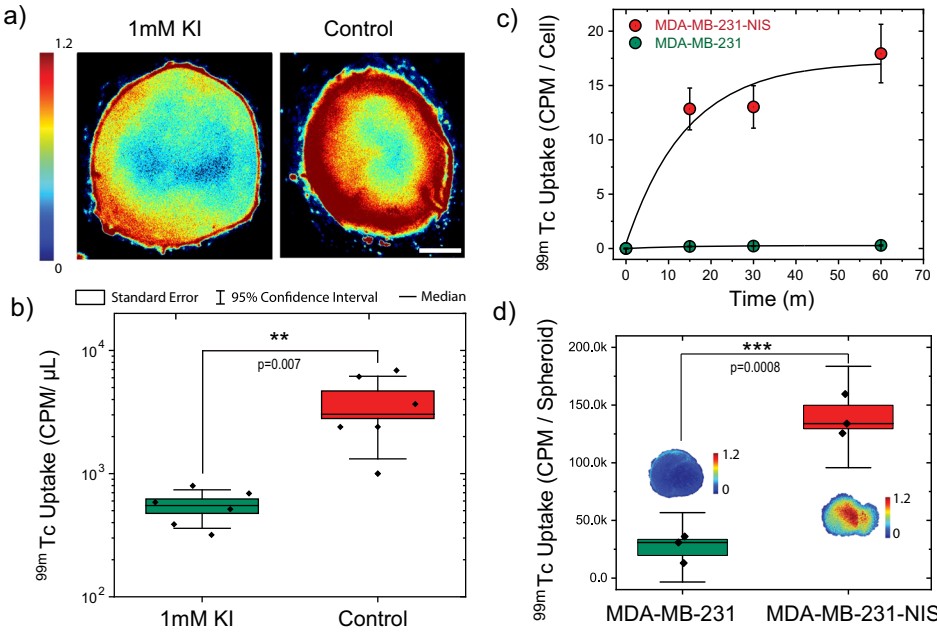

**Fig. 6 Functional characterization of NIS activity in tumor organoids derived from papillary thyroid carcinoma patients. a** oPEM imaging of pertechnetate ($^{99m}TcO_4^-$) uptake by tumor organoids with (left) or without (right) competitive inhibition by 1 mM potassium iodide (KI). Scale bar: 200 μm. **b** A nearly 10-fold difference in $^{99m}TcO_4^-$ uptake is observed between the two experimental conditions ($n = 6$ independent tumor organoids). Data are shown as mean ± S.E.M. (box), median (horizontal line), and 95% confidence interval (CI; whiskers). **c** $^{99m}TcO_4^-$ uptake kinetics of NIS-expressing and wild-type MDA-MB-231 cells. Error bars correspond to the standard error from $n = 3$ independent measurements. **d** Gamma counting and oPEM images of $^{99m}TcO_4^-$ uptake by 3D multicellular spheroids (10,000 cells/spheroid) made from NIS-expressing and wild-type MDA-MB-231 cells ($n = 3$ independent spheroids). Data are shown as mean ± S.E.M. (box), median (horizontal line), and 95% CI (whiskers). Statistical significance was assessed using a two-tailed unpaired $t$ test (**b**, **d**). CPM: counts per minute. The color bar shows radioactivity (Bq/pixel).

fashion using PET biomarkers that are also used as part of standard clinical workflow. The method could be easily adopted in the clinic as oPEM is easy to operate for clinical staff who are skilled in performing histological examination under a standard microscope. In particular, it could be useful in clinical decision-making for those patients who fail the standard initial treatment. As the turnaround time of the tumor organoids is shorter than typical treatment cycles, oPEM could be used to predict the best-personalized regimen for the second-line treatments. (Supplementary Fig. 11). This capability could also be used as part of co-clinical trials, which use organoid models derived from patients enrolled in clinical trials[40]. Future studies are warranted to successfully translate this technology for its intended clinical applications.

PET is an outstanding tool for imaging molecular processes in vivo. Its picomolar-concentration sensitivity is unequaled in the clinical arena and countless radiotracers are available to image the unique hallmarks of cancer[46,47]. A new generation of PET tracers is now entering clinical use that can image cancer-specific targets such as bombesin receptors, prostate-specific membrane antigen, somatostatin receptor, $\alpha_v\beta_6$ integrin, and fibroblast activation protein[48–50]. Some of these emerging tracers are unique in that they do not target tumor cells but, rather, specific alterations of the tumor microenvironment. For instance, tumor-specific markers are expressed on the neovasculature of tumors[51]. Other markers are displayed by tumor-associated fibroblasts. In addition, tracers have been developed to assess the activity of tumor-infiltrating lymphocytes[52], such as $^{18}$F-AraG which is being used in clinical trials to measure T-cell activation during immunotherapy[53]. Tumor organoids contain a broad spectrum of cells including immune and stromal cells, and thus can be used as a model to evaluate and screen advanced PET tracers. For instance, we found that organoids can be imaged with

both FDG and $^{18}$F-AraG to assess tumor-killing and T-cell activation, respectively (Supplementary Fig. 12). Therefore, tumor organoids could become a valuable model to test and improve oncologic PET tracers.

It should be noted that RLM can image different types of charged particles over a wide range of energies, and thus it is not limited to any specific radioisotope such as $^{18}$F or $^{99m}$Tc. Commonly used PET radioisotopes include $^{18}$F, $^{68}$Ga, $^{64}$Cu, $^{124}$I, and $^{11}$C, all of which can be easily imaged with similarly high resolution. The different positron range of these isotopes minimally affects spatial resolution because only a small section of the scintillator is seen in focus by the microscope. In addition, therapeutic alpha and beta radiation could be imaged for microdosimetric evaluation of radiopharmaceutical therapy and response assessment in organoid tumor models. Organoid models of tumors or healthy organs may, therefore, provide a compelling new platform for the assessment of theranostic regimens.

There are several developments that could further enhance organoid imaging using oPEM. First, as the current study is limited by the small number of patients, a larger study would be necessary to advance this method towards its intended application. Second, oPEM should be compared with current approaches in terms of its ability to accurately predict in vivo responses. One potential advantage of oPEM is that it measures a quantitative biomarker (the rate of uptake of a radiopharmaceutical) in a way that is testable subsequently using in vivo PET imaging. A third potential improvement would be to develop a high-throughput technology to measure FDG uptake in hundreds of organoids for large-scale drug screening and other applications. For instance, one possible approach would be to build 96-well plates with a scintillator bottom and automate stage movements for faster imaging. Similar to the various prospective drug screening platforms that have been developed using 2D cell culture, spheroid

culture, mouse & zebrafish avatars, and patient-specific induced pluripotent stem cells[13,17,54–56], high-throughput organoid screening would allow testing a range of treatment options for individual patients on the basis of FDG uptake, but with a better trade-off between accuracy, ease of use, turnaround time, and cost-effectiveness.

In conclusion, we demonstrated the use of oPEM for imaging tumor organoids with clinical radiotracers. Given the expanding use of organoid models in research, we envision a number of emerging applications for this imaging technique. For instance, FDG-oPEM could be used to select suitable therapies for individual patients on the basis of FDG imaging of patient-matched organoids. Second, the approach could be used as an endpoint to screen drug candidates and new PET radiotracers using diverse panels of organoids. Third, it could be used to functionally validate newly derived tumor organoids and show that these models truly reflect the physiological state of the tumor from which they were derived. Finally, oPEM and PET, together, could provide unified image-derived biomarkers to integrate, in real-time, the data from the preclinical and clinical arms of the co-clinical trial. The development of small-animal PET scanners 15 years ago has demonstrated the utility of developing preclinical imaging tools that have strong relevance to the clinical workflow. Accordingly, oPEM could become a standard technology for imaging tumor and normal tissue organoids using PET tracers, thereby expanding the use of organoids models for translational research.

## Methods

**Human specimen collection.** Freshly resected primary and metastatic tumor tissues were obtained through the Stanford Tissue Bank from patients undergoing surgical resection at Stanford University Medical Center. All experiments utilizing human material and associated imaging, including positron emission tomography–computed tomography, were approved by the Institutional Review Board at Stanford University. Written informed consent for research was obtained from donors before tissue acquisition. The PET/CT data were obtained from either Stanford PACS or electronic health record (Epic) and DICOM images were visualized and processed in Osirix 12.0.3. The patients were from both sexes, between 29 and 70 years of age, and weighed between 59 and 107 kg.

**Organoid culture.** Excess fat or muscle tissue from human tumor tissue was removed to enrich the samples for epithelial cells. Tumor tissues were minced finely (~2–3 mm) on ice, washed twice in DMEM/F-12 (Invitrogen) containing 1× Primocin (InvivoGen), and followed with ACK Lysing Buffer (Thermo Fisher Scientific) to remove red blood cell contamination. Minced tissues were washed again with DMEM/F-12 media and resuspended with cold BME (Cultrex® reduced growth factor basement membrane matrix, type 2, Trevigen). Droplets of ~40 µl were plated on the bottom of pre-heated 24-well culture plates (E&K Scientific). After plating, culture plates were incubated at 37 °C for 30 minutes to let the BME matrix polymerize. Organoid culture medium (EN medium) containing DMEM/F-12 supplemented with 10% Noggin-conditioned medium (HEK293, ATCC), Nicotinamide (10 mM, Sigma), N-Acetylcysteine (1 mM, Sigma), B-27 without vitamin A (1×, Invitrogen), Pen–Strep (1×, Invitrogen), and EGF (50 ng/mL, R&D Systems) was added. For lung, kidney, and colon organoids, WERN media containing EGF, Noggin, RSPO1, and Wnt3a were used following a previous protocol[5].

The medium was changed every 2–3 days and organoids were passaged every 2–4 weeks by dissociation with 1 ml of TrypLE Express (Life Technologies) at 37 °C for 5 min. Tissue suspension was sheared using 1 ml pipette. Digestion was monitored closely to prevent excess incubation in trypsin. Organoid pellets obtained after centrifugation at 500 × g were washed with DMEM/F-12 media (containing 10% FBS and Pen–Strep) and replated at a 1:3 split with BME matrigel. For Cisplatin (Millipore Sigma) treatment, 1–10 µM of Cisplatin was added to the media for 24 h with DMSO as a control. For cryopreservation of organoid tissues, freezing media (90% FBS containing 10% DMSO) was used with a standard freezing protocol. The organoids shown in Fig. 2e and supplementary fig. 12 were cultured using an air-liquid interface (ALI) method to better retain the tumor-infiltrating immune cells as described below and in a previous study by Neal J et al.[5].

**ALI organoid culture.** To culture organoids in ALI system, minced tissue pellets were resuspended in 1 mL of type I collagen gel (Trevigen), and layered on top of 1 mL of pre-reconstructed collagen within a 30 mm diameter/0.4 µm inner transwell dish (Millicell). The inner transwell dish was placed into an outer 60 mm tissue culture dish containing 1.5 mL of EN medium, and the lid of the outer dish was replaced as previously described[57]. Organoids were passaged every 7–14 days by dissociation with 300 units/ml collagenase IV (Worthington) at 37 °C for 30 minutes, followed by three 5-minute washes with 100% FBS and propagation at a 1:4 ratio into new ALI collagen gels. The growth medium was replaced once every week. To examine the T-cell activation using 18F-AraG in the ALI system, the organoid medium was supplemented with 1000 IU/ml recombinant human IL-2 (Peprotech) or 10 µg/mL control mouse IgG2b (BE0086, Bioxcell) for 7 days.

**Histology and immunofluorescence analysis.** Organoids were fixed in 4% paraformaldehyde, paraffin-embedded, and sectioned by Stanford Human Pathology and Histology Service Center. Sections (4–5 µm) were deparaffinized and stained with hematoxylin and eosin (H&E). For immunofluorescence staining, the following antibodies were used. Primary antibodies: anti-E-cadherin (BD, 610181, 1:1000), anti-vimentin (Millipore, AB1620, 1:1000), and anti-CD3 (Dako, A0452, 1:100). All secondary antibodies were used at 1:500 and were: IgG (H + L)-Texas Red (goat anti-mouse; Thermo Fisher, # T-862, 1:1000), IgY (H + L)-AF488 (goat anti-chicken; Thermo Fisher, # A-11039, 1:1000), and IgG (H + L)-Cy5 (goat anti-rabbit; Thermo Fisher, # A10523, 1:1000). All images were captured on a Zeiss Axio-Imager Z1 with ApoTome attachment.

**RNA in situ hybridization.** RNA expression was performed on 5 µm sections using the RNAscope® 2.5 HD Duplex Detection Kit (Cat. No. 322435, Advanced Cell Diagnostics) and probes for GLUT1, CD44, and Ki67 (Advanced Cell Diagnostics). Tissue sections were pre-treated with heat and protease prior to hybridization of the target probe. Preamplifier, amplifier, and alkaline phosphatase labeled oligos were sequentially hybridized followed by the application of a chromogenic substrate to produce blue and red punctate staining. Sections were counterstained with Gill's hematoxylin (Sigma-Aldrich) and mounted with Ecomount (Biocare Medical).

**Radiolabeling and sample preparation.** Medical-grade FDG and 18F-AraG were produced at the Stanford radiochemistry facility using an on-site cyclotron. 99mTc-pertechnetate was eluted from a clinical generator. For imaging of glucose metabolism using combined fluorescence and RLM, organoids were incubated in a glucose-free DMEM medium for 30 min before the addition of FDG. Standard DMEM was used for 99mTc and 18F-AraG experiments. The organoids were incubated with the respective radiotracers at a concentration of 1 mCi/ml for 1–2 h at 37 °C followed by washing with phosphate-buffered saline (PBS) thrice with gentle rocking. The hydrogel was gently disrupted, and the organoids were picked using a pipette tip and transferred to PBS for final washing. The organoids were washed thrice in PBS to remove all unbound FDG. Individual organoids were carefully collected and gently placed on a 0.5 mm thick CdWO₄ or a 0.2 mm thick GAGG:Ce scintillator plate (Kinheng crystal material, Shanghai), which itself was mounted onto a standard glass coverslip (0.1 mm thick, Fisher Scientific). The specimens were then promptly placed on the microscope stage for multimodal imaging. The specimen remained hydrated during the short imaging time. For in situ imaging of organoids inside the matrigel, washing steps were performed at 37 °C and extended to 30 min to ensure sufficient signal-to-background. A 0.2 mm thick GAGG:Ce scintillator was placed on the top of the matrigel dome and gently compressed to form a thin matrigel film. The 24-well plate was then mounted on the microscope stage for imaging organoids inside the matrigel.

**Instrumentation and imaging.** Several scintillator materials can be used for RLM. CdWO₄ has a moderately high light yield (12,000–15,000 photon/MeV), high effective atomic number (Zeff = 64), high density (7.9 g/cm³), and no significant afterglow. GAGG:Ce (Zeff = 54.4, density=6.6 g/cm³) has a higher light yield (~57,000 photon/MeV) in the yellow region of the spectrum (560 nm). The set-up was mounted on a custom-built wide-field microscope equipped with a short focal tube lens (×4/0.2 NA; Nikon, CFI Plan Apochromat λ), ×20/0.75 NA air objective (Nikon, CFI Plan Apochromat λ), and deep-cooled EMCCD (Hamamatsu Photonics, ImagEM C9100-13)[23]. Data were collected using MicroManager 1.4.22 software.

Brightfield images were acquired with no EM gain. For RLM imaging, images were taken with a ×20 objective, an exposure time of 10–300 s, an EM gain of 600/1200, and no pixel binning. We used the brightfield mode to set the microscope into focus. Optimal radioluminescence focus was achieved when the organoids displayed sharp positive contrast in the corresponding brightfield image. For fluorescence microscopy, we used 387 mm/447 mm filter set (Semrock, filter ref: DAPI-1160B-000) for Hoechst 33342 (live cell marker) and 469 nm/525 nm filter set (Thorlabs, filter ref: MDF-GFP1) for 2-NBDG or SYTOX green (dead cell marker).

ImageJ 1.52a (NIH) and MATLAB (version R2020b) were used to process and analyze the raw images. Digital images were reconstructed using ORBIT 1.11. OriginLab 2019b was used for graphical analysis.

**Quantification of FDG from oPEM.** To quantify the FDG dose inside organoids, the EMCCD radioluminescence signal was calibrated with a known amount of radioactivity. Radioluminescence signal from a microdroplet of FDG with known activity was measured with a 5 min exposure time. Images were acquired over 8 h

while the radioactivity gradually decayed down to a very low value. The intensity of the signal in a region of interest was plotted against time and compared with the 110-min half-life decay curve of $^{18}$F. The camera signal integrated over the microdroplet area was scaled linearly to obtain the spatial concentration of radioactivity in Bq/pixel.

**Influx rate computation**. To properly account for the time-varying concentration of FDG available to tumor cells, we calculated the net rate of FDG influx, noted $K_i$, into patient and organoid tumors. The parameter $K_i$ was calculated from a simplified Patlak model (assuming zero-intercept) with the following equation:

$$C_{\text{ROI}} = K_i \int_0^T C_p(t)dt \qquad (1)$$

where $C_{\text{ROI}}$ is the concentration of the tracer in the region of interest and $C_p$ is the concentration of tracer in the plasma. $C_p$ was assumed to be constant in the tumor organoids, which were incubated in a fixed concentration of FDG for a known duration. For patients, the estimation of $C_p$ is more complex and ideally would necessitate serial arterial sampling to determine a time-dependent input function. To overcome this challenge, we used a standardized input function (SIF) derived from a population of 101 patients[36] to compute a patient-specific input function $C_p$:

$$C_p(t) = \frac{\text{ID}}{\text{EDV}} \times \text{SIF}(t) \qquad (2)$$

$$\text{EDV} = 39 \times H^{0.8} \times W^{0.35} \qquad (3)$$

where ID is the injected FDG dose, EDV is the estimated distribution volume of FDG in a patient of height $H$ and weight $W$ estimated using Eq. (3), and SIF(t) is the function provided by Shiozaki et al.[36].

The FDG concentration $C_{\text{ROI}}$ was estimated from the maximum SUV value in the patient tumors. In organoids, $C_{\text{ROI}}$ was calculated by measuring the total radioactivity in a gamma counter (Hidex) and dividing by individual organoid volume. The volumes were estimated from the area measured using optical microscopy, assuming a spherical shape of those organoids. A calibration grid (Thorlabs, cat. no. R1L3S3P) was used for the accurate estimation of the organoid area (cross-section). To strengthen our findings, quantification of individual organoids was performed both using oPEM imaging and gamma counting, however, different samples from the same source were used for both tests.

**Statistics and reproducibility**. FDG-oPEM imaging was performed in >50 tumor organoids from six different patients with similar image quality and spatial resolution. Tc99m imaging was performed >10 tumor organoids with similar imaging results. All experiments were performed with organoids form at least two independent cultures. Except Fig. 5, at least three biological replicates were used in all experiments as indicated in the figure legends. Statistical differences between two treatment groups were calculated using unpaired two-tailed student's $t$ tests. The level of significance is indicated as follows: $*P < 0.05$, $**P < 0.01$, $***P < 0.001$.

**Reporting summary**. Further information on research design is available in the Nature Research Reporting Summary linked to this article.

## Data availability
The main data supporting the results in this study are available within the paper and its Supplementary Information. The raw EMCCD image files generated in this study have been deposited in the Dryad open-access repository[58].

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

## Acknowledgements

This work was funded in part by NIH grants R01CA186275, U19AI116484, U01DK085527, U01CA217851, U01CA176299, U01DE025188, and K00CA223019. This project was also funded in part by a grant from the Stanford-Tuebingen Collaboration on Imaging Immunotherapy. The authors thank the Stanford radiochemistry facility for providing FDG, 18F-AraG and 99mTc. They also wish to gratefully acknowledge assistance from Dr. Tae Jin Kim for assistance using the RLM and Dr. Jia Wu for providing PET images for illustration purposes. NIS-expressing MDA-MB-231 cells were a generous gift from Dr. Irene Wapnir. Figure 1 and S11 were created using BioRender.com.

## Author contributions

S.K. performed multimodal PEM imaging, developed new methodologies, analyzed data, and wrote the initial manuscript. J.H.S. performed organoid culture and histology of all patient-derived tumor samples. J.B.S. and J.E.N. provided surgical tumor samples. In addition, J.B.S. supervised organoid development. N.C. performed a comparative analysis between fresh tumor and organoid culture under C.K.'s supervision. V.F. processed and analyzed clinical PET/CT data for $k_i$ calculation. G.P. conceived the research idea, developed $k_i$ analysis method, wrote the manuscript, and supervised the study.

## Competing interests

The authors declare no competing interests.
