## [Peer Review File · Nature Communications]

REVIEWERS' COMMENTS

Reviewer #1 (Remarks to the Author):

The authors responded again well to my prior comments and it is a nice methodological paper.

The only experiment I would suggest they contemplate is to incubate individual spheroids with FDG, and then INDIVIDUALLY measure uptake in the spheroid and then place them directly (in a drop of medium) onto the scintillator, Correlate the uptake with the signal. And, if possible, also correlate with the in vivo signal from the same tumor (patient PET scan.)

Comment: The authors responded again well to my prior comments and it is a nice methodological paper.

Response: We are grateful to the reviewer for reviewing our manuscript and helping to improve it significantly.

Comment: The only experiment I would suggest they contemplate is to incubate individual spheroids with FDG, and then INDIVIDUALLY measure uptake in the spheroid and then place them directly (in a drop of medium) onto the scintillator, Correlate the uptake with the signal. And, if possible, also correlate with the in vivo signal from the same tumor (patient PET scan.)

Response: We appreciate this suggestion from the reviewer. We think our protocol was very similar to the one suggested by the reviewer. Culturing a single organoid is challenging, as many organoids grow independently in a matrigel dome in our current setup. However, we do pick individual organoids from the matrigel dome after incubation with FDG and place them on the top of scintillator. Apart from few in-situ imaging inside matrigel dome, all organoids has been individually imaged in our custom built setup and/or individually measured using gamma counting.